# The Role of Social Media in Health Safety Evaluation of a Tourism Destination throughout the Travel Planning Process

**Claudia-Elena Țuclea [1,*], Diana-Maria Vrânceanu [2,*] and Carmen-Eugenia Năstase [3]**

[1] Department of Tourism and Geography, Faculty of Business and Tourism, Bucharest University of Economic Studies, 010404 Bucharest, Romania

[2] Department of Marketing, Faculty of Marketing, Bucharest University of Economic Studies, 010404 Bucharest, Romania

[3] Department of Economics, Business Administration and Informatic Economics, Faculty of Economics and Public Administration, Ștefan cel Mare University of Suceava; 720229 Suceava, Romania; carmen.nastase@usm.ro

[*] Correspondence: claudia.tuclea@com.ase.ro (C.-E.Ț.); diana.vranceanu@mk.ase.ro (D.-M.V.)

**Abstract:** This research aims at identifying the role of social media in evaluating the attractiveness of a tourism destination, with special emphasis on the health safety of the destination. Consistent with this objective, a survey has been carried out on a sample of 675 Romanian social media users. The research results led to the development of a model based on structural equation modeling. The model includes nine latent variables that were structured taking into account different behavioral aspects related to the role social media has in travel planning, as well as for evaluating the health safety of a tourism destination. The main findings suggest that the trust in social media for tourism information made people become more interested in communicating through this means and to consider it more useful throughout the travel planning process. When choosing a travel destination, the more involved a tourist is in the decision making process, the greater the attention they pay to social media. The perceived usefulness of social media in travel planning has a significant influence on intentions to choose a tourism destination. As the importance assigned to the health safety of tourism destination increases, social media plays a more active role in travel by creating trust in this means in order to obtain sanitary safety information. People that intend to use social media for finding information on the health safety of a tourism destination are more likely to choose that destination for their vacation. The managerial implications of this paper regard the communication strategies adopted by tourism services suppliers or by some public authorities aimed at stimulating an efficient usage of social media so as to increase the buying intentions for tourism destinations.

**Keywords:** social media; sanitary safety; social media for tourism; health-related crises; social media marketing; mobile marketing approach; travel destination

---

## 1. Introduction

It is a common fact that social media has become an indispensable component of contemporary human life. A significant part of communication takes place through social media [1]. Worldwide, 63% of internet users claim they are constantly connected online and spend daily, on average, 2 h and 24 min on social networks and messaging applications [2].

The widespread use of the Internet and the rapid progress of technology have changed almost all industries, tourism being one of the deeply reconfigured sectors [3]. In this context, social media has acquired an important role in tourism activity, both among tourists and providers [4–6]. On the one

hand, social media allows tourists to access and use information [7], but also to generate content [8]. Tourists are looking for the most comprehensive information about the destination where they choose to spend their vacation, with the intention to minimize risk and uncertainty about the quality of services and safety [9]. They gather information about destinations, transport and accommodation offers, compare prices and services and search through photos and videos from these destinations [10].

On the other hand, tourism providers depend on the most convincing and attractive information about their offers. For this reason, they are directly interested in the content that tourists generate and it would be to their benefit to encourage tourists to share their impressions of the holiday destination [11]. As a result, tourism companies can take on a social context mobile marketing approach (SoCoMo) [12] and even try to develop an image of the destination on social media that is based on an emotional appeal [13]. Thus, destination management organizations have understood the need to share more visual content on their official social media platforms [14]. This has become even more important, as tourist behavior can be influenced by both direct and intentional recommendations from social media friends, as well as unintended actions related to brand generated by social media friends [15].

At the same time, tourism is a sector severely affected by various types of crises [16], including both financial [17] and natural disasters [18]. Health crises have a strong negative impact on tourism [19,20].

It is expected that the informative role of social media will acquire a particular relevance during health-related crises such as the current crisis, generated by the pandemic caused by SARS - CoV-2. Consequently, the present research aims to identify the role of social media in vacation planning, with a particular emphasis on information related to health safety of the destination and its importance in the travel planning process.

Further, the paper is organized as follows: Section 2 reviews the literature related to the role of social media in the travel planning process and also the importance of social media information in creating the perception of the health safety of the holiday destination. In this section, the hypotheses are also presented. The research methodology and data analysis are introduced in Section 3, while the results and discussions are detailed in Section 4. The theoretical and practical implications of the findings are discussed in the last section. The study's limitations are outlined and directions for future research are suggested.

## 2. Literature Review and Hypotheses

Social media has become the way through which people get involved in online activities, especially in order to create and share information from anywhere at any time. This fact has, however, less desired consequences for users, as it has become difficult to identify relevant information [21,22]. From another perspective, the quality of generated content can widely vary, from high value content to manipulation, fake news or spam, which opens up the possibility to contaminate social networks with unwanted and unsafe content. Thus, there is a problem evaluating the credibility of information, which influences the understanding of events, exposing users to risks [22]. Therefore, users' trust in social media is essential in the process of searching for relevant and reliable information, and trust in social media represents a research topic of growing importance and useful implications [21,23]. Some authors have shown that young people trust social media information and use it in the planning of a holiday [24], and others have demonstrated that there are some small age differences in social media adoption and frequent similarities between younger and older tourists regarding the trustworthiness of social media channels [25].

In this regard, the present study puts forward a first hypothesis:

**Hypothesis 1 (H1).** *Trust in social media for tourism influences the degree of involvement in social media on tourism issues.*

The successful involvement of tourists on social networks requires increased attention to the content, format and timing of posts, as well as the expression of feelings [26]. These aspects lead to the

existence of digital skills that, if they do not yet have, then many tourists do try to acquire. Correa [27] showed that more educated and skilled people tend to use Facebook for informational and mobilizing purposes. Children also teach their parents how to use digital media, a process more common among women and people with lower socioeconomic status and also associated with less authoritarian parents and more relaxed relations between parents and children [28,29]. Other authors have shown that tourists are not only passive consumers of information shared on social media, but are also initiators and distributors of information [30]. All these activities carried out on social media are based on a certain level of skills of using social media, which leads to the second hypothesis of this study:

**Hypothesis 2 (H2).** *Social media skills influence the degree of involvement in social media on tourism issues.*

Many researchers have already written about the significant role that social media plays in many aspects of tourism activity [31]. A particular emphasis was placed on searching for information and decision making behaviors, on the role of social media in promoting tourism and in identifying best practices in interaction with consumers [32]. Other authors [33] have shown that social media plays an important role when people make travel plans. Studies have revealed that tourists use social media at all stages of travel planning, considering it a reliable source of information [34,35]. Uysal, Perdue and Sirgy [36] noted that many tourists consider social networks as a reliable source of information that could help them search and plan their journey, primarily due to the fact that materials accessed through social networks present the current state of the destination that interests them. Through social media platforms, social media users can find essential information about the desired destination [36,37]. These conclusions lead us to the assumption of the third hypothesis:

**Hypothesis 3 (H3).** *The degree of involvement in social media on tourism issues influences the perceived usefulness of social media in the travel planning process.*

In addition to the degree of involvement in social media on tourism issues, tourists using social media platforms are also concerned with the process of travel destination and tourist services choice. Choosing a holiday destination is a complex process, with many components and influenced by a number of psychological and non-psychological variables [38]. The attitude of tourists towards a destination and the perceived control over the trip are significantly influenced by the secondary information that gives tourists the opinions of other people about potential destinations [39]. A significant source of secondary information comes from word-of-mouth (WOM) [40]. Park and Kim's research [41] indicated that a strongly involved tourist relies on both his own experience and the recommendations of others to make a travel decision. Other studies have highlighted the transition to eWOM, showing that in tourism, online resources, including eWOM, have facilitated the search for travel information [42,43]. eWOM also contributes to the evaluation of choices regarding destination tourist services, especially at the micro-destination level [44,45].

Analyzing the influence that social media has on the choice of destination, Tham et al. [46] argue that studies highlighting such an influence were conducted in contexts in which tourists were prone to be influenced (contexts were selected precisely on the basis of social media presence or influence). Corroborating these issues, hypothesis H4 is submitted:

**Hypothesis 4 (H4).** *The degree of the user's involvement in choosing tourist services influences the perceived usefulness of social media in the travel planning process.*

eWOM communication has also gained increasing attention, as it has started to influence more and more tourists in choosing a destination [47] and with their purchase intention [48,49]. Abubakar stated that eWOM is positively related to destination trust [50]. Among the important factors that make a destination reliable for tourists are health and hygiene. Health care and sanitation are important elements for attracting tourists, representing a guarantee in disease prevention [51]. The health risks

inherent, to a certain extent, during the journey and stay in a destination, can influence the tourists' perceptions about the risk and, therefore, the choice of destination and their tourist behavior [52]. Regarding this idea, some authors [53] analyzed how US tourists perceived the health risks associated with travel and how they prepared for their international travel. Health risks lead tourists to look for information [52], in order to help them choose the destination, improve the quality of their trip and reduce the risks and uncertainties related to the trip experience [54]. In this context, this study puts forward a fifth hypothesis:

**Hypothesis 5 (H5).** *There is a direct link between the degree of involvement in the choice of tourist services and the importance of health safety in choosing the travel destination.*

Social media is one of the main sources of information for tourists [55]. The decision of tourists about a holiday destination or a trip is largely influenced by the recommendations of relatives or friends, online recommendations, as well as comments and information provided by social media platforms [56]. The final decision on the choice of destination is most often based on information from eWOM [57]. Although the issue of the credibility of social media as a source of information is widely debated, many studies show that social media travel and tourism information sources are more trustworthy than other sources [56]. A recent study reiterates that trust in information sources stimulates the intention to buy [58], while online reviews of travel and user-generated content are appreciated by social media users as more reliable information than that provided by tourism organizations [59], especially if they are created and published by real independent people who describe real experiences [33,59,60]. Potential tourists are looking for online reviews because tourist services are not accessible until the time of consumption and thus the risk and uncertainty associated with them are increased [61]. The concern for reducing these risks and uncertainties becomes even more important when it comes to health security in the tourist destination [19]. This is highlighted by most studies on the impact that the risk perceived by tourists on the health security of a destination has had on the number of international arrivals in those destinations [16].

Considering, on the one hand, the importance that tourists confer to health safety in choosing a tourist destination, and, on the other hand, the degree of tourists trust in social media information on this issue, as well as the need for information in making the final decision, we formulate the following two hypotheses of the present study:

**Hypothesis 6 (H6).** *The importance given to health safety in choosing a travel destination influences the degree of trust in social media towards information on the health safety of a destination.*

**Hypothesis 7 (H7).** *The degree of trust in social media towards information on the health safety of the destination influences the intention to use social media to determine the health safety of a travel destination.*

In all health related crises, the media has played an important role in the perceptions held by tourists of the associated risk [19], and the perceptions of the risk have proved to be an important vector for information search on social media [62]. Frequency of travel has also been shown to influence the intention of using social media in the event of a travel crisis [63].

Consistent with the above, we formulate the following two hypotheses of the study:

**Hypothesis 8 (H8).** *The perceived usefulness of social media in the travel planning process influences the intention to use social media for information on the health safety of a destination.*

**Hypothesis 9 (H9).** *The intention to use social media regarding the health safety of the travel destination influences the intention to purchase a tourist offer in a certain destination.*

Nowadays, there are social media applications installed on every smart device and they are used as a tool to find more information about traveling [33]. This information can be very useful for potential tourists and can be personalized [64]. The usefulness of social media for tourists is provided by several factors, such as: source credibility, information reliability, enjoyment and perceived value [65]. The source credibility is an important factor in decision making [66] and has a positive influence on the purchase intention [65]. Reliability of information is considered an important attribute that influences the perceived value of social media [67]. The ease of the search process combined with the reliability of the information help tourists evaluate a destination and have a positive influence on the purchase intention [65]. Enjoyment plays a significant role in the acceptance of technology [68] and determines the frequency of use of a computer because this can generate a pleasant feeling. The pleasant feelings generated by the use of social media applications encourage tourists not only to search for information about travel destinations, but also to interact with others. Tourists interact with each other by sharing photos or videos [67]. The theory of perceived values was also adopted in connection with the choice of travel destination. This shows high levels of influence on the future intention of travelers to discover new destinations or to return to destinations already visited [69]. Tourists evaluate the information from social media based on their perceptions of what they want to achieve and what they are ready to sacrifice [70].

Taking into consideration all these factors that define the utility of social media perceived by tourists, we propose the last hypothesis of this study:

**Hypothesis 10 (H10).** *The perceived usefulness of social media in the travel planning process influences the intention to purchase a travel destination.*

The proposed model is presented in Figure 1:

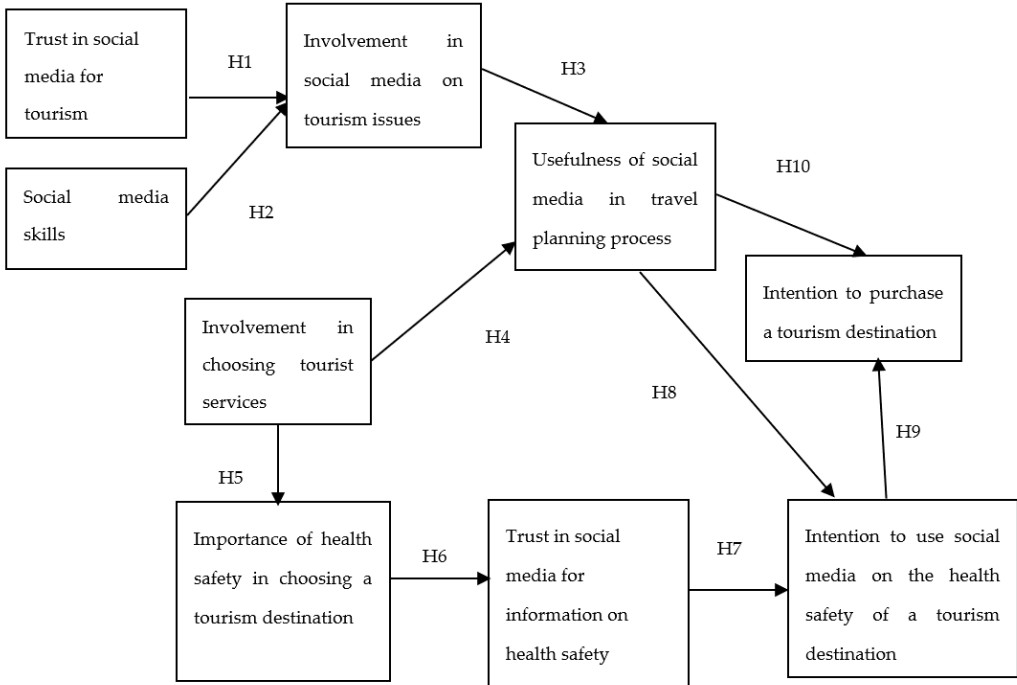

**Figure 1.** The conceptual model.

## 3. Research Methodology and Data Analysis

The research method consisted of a survey at the national level where the data was collected during May-June 2020 with the help of an online platform. The data was processed using Excel, and the data analysis was based on the Structural Equation Modeling approach performed through

the WarpPLS 5.0 software. The statistical population was represented by Romanian users of social media, the sample size being 675 persons. The sampling error was ±3.77% for 95% level of confidence (t = 1.96). The sampling method was stratified, using as stratification variables: gender, age, education and income. The sample structure is presented in Table 1:

**Table 1.** The sample structure.

| Variable | | Frequency | Percent |
|---|---|---|---|
| Gender | Men | 326 | 48.3 |
| | Women | 349 | 51.7 |
| Age | 18–24 | 132 | 19.6 |
| | 25–40 | 211 | 31.3 |
| | 41–54 | 215 | 31.9 |
| | 55 and over 55 | 117 | 17.3 |
| Education | Undergraduates | 47 | 7.0 |
| | High school | 218 | 32.3 |
| | Post high school | 73 | 10.8 |
| | University | 280 | 41.5 |
| | Post university | 57 | 8.4 |
| Income | Under 1200 RON | 91 | 13.5 |
| | 1200–2000 RON | 113 | 16.7 |
| | 2001–3000 RON | 180 | 26.7 |
| | 3001–4000 RON | 158 | 23.4 |
| | 4001–5000 RON | 67 | 9.9 |
| | Over 5000 RON | 66 | 9.8 |
| **Total** | | 675 | 100 |

For data analysis, the structural equation modeling procedure was performed in order to evaluate the relationships between the latent variables taken into account in the designed model. The latent variables (Table 2) were measured by using a semantic differential scale, from 1-Not at all to 5-Very much, and all of them were reflective. In order to increase the accuracy of the data, the reliability and validity of the construct were assessed.

**Table 2.** Latent variables.

| Variable | Items | Sources (Adapted from) |
|---|---|---|
| Social media skills (Sk_sm) | Sk_sm 1: You consider yourself an experienced social media user | Liu, Y.; Bakici, T. [71] Lee, H.; Place, K.R.; Smith, B.G. [72] |
| | Sk_sm 2: You are familiar with the use of social media | |
| | Sk_sm 3: You trust social media information | |
| | Sk_sm 4: You have no difficulties in using different social media platforms | |
| | Sk_sm 5: You make various decisions in daily life based on what you find using social media (e.g., what to cook, what restaurant to go to, what show to watch, etc.) | |
| Trust in social media for tourism (Tr_sm) | Tr_sm1: You trust the opinions of visitors to some tourist attractions posted on social media | Berhanu, K.; Sahil, R. [56] Narangajavana, Y.; Callarisa-Fiol, L.J.; Moliner-Tena, M. Á.; Rodríguez-Artola, R.M.; Sánchez-García, J. [4] |
| | Tr_sm1: You trust the reviews and comments of friends posted on social media regarding a travel destination | |

**Table 2.** *Cont.*

| Variable | Items | Sources (Adapted from) |
|---|---|---|
| Involvement in social media on tourism issues (Inv_sm) | Inv_sm1: You share comments on social media about a visited destination | Oliveira, T.; Araujo, B.; Tam, C. [73] Kang, M.; Schuett, M.A. [55] Jiang, Z.; Chan, J.; Bernard, C.Y.T.; Chua, W.S. [74] |
| | Inv_sm2: You share photo content on social media about a visited destination | |
| | Inv_sm3: You share video content on social media about a visited destination | |
| | Inv_sm4: You share reviews on specialized tourism sites regarding accommodation units, restaurants, tourist attractions, etc. | |
| | Inv_sm5: You are influenced by your social media audience to share content about your travel destination | |
| Involvement in choosing tourist services (Inv_t) | Inv_t1: You consider yourself an experienced tourist | Sharma, V.M.; Klein, A. [75] Zaichkowsky, J.L. [76] Suhartanto, D.; Dean, D.; Nansuri, R.; Triyuni, N.N. [77] |
| | Inv_t2: You consider yourself an exigent tourist | |
| | Inv_t3: As a tourist, you are always looking for new experiences, adventure, variety | |
| | Inv_t4: You are concerned with responsible tourism (which does not affect nature and local communities) | |
| Importance of health safety in choosing the travel destination (Hlth_s) | Hlth_s1: Compared to previous holidays, the health safety of the destination and the additional hygiene rules will count more in the choice of future vacations (after the Covid-19 crisis) | Wall, G. [78] Okuyama, T. [79] Chilton, A. [80] Rosen, E. [81] |
| | Hlth_s2: In choosing a travel destination, you are willing to give up certain facilities if it leads to an increased health safety | |
| | Hlth_s3: You would be willing to pay more if you obtain additional guarantees regarding the health safety of the travel destination | |
| Intention to use social media regarding the health safety of a travel destination (Int_smh) | Int_smh1: Various tourism blogs / vlogs | Harrigan, P.; Evers, U.; Miles, M.; Daly, T. [82] |
| | Int_smh2: Social media accounts of friends / acquaintances | |
| | Int_smh3: Accounts on social networks of influencers in the tourism area | |
| | Int_smh4: Forums with tourists' opinions | |
| | Int_smh5: Official sites from social networks of tourism service providers | |
| | Int_smh6: Various podcasts | |
| | Int_smh7: Various platforms for distributing video content (e.g., Youtube) | |
| | Int_smh8: Various platforms for distributing photo content | |
| | Int_smh9: Tourism oriented sites (e.g., TripAdvisor) | |
| Usefulness of social media in the travel planning process (Use_sm) | Use_sm1: You use social media information to plan your holiday | Kang, M.; Schuett, M.A. [55] |
| | Use_sm2: You use social media information to practice responsible tourism (a form of tourism that does not affect nature and local communities) | |
| | Use_sm3: Social media information influences your behavior in the travel destination | |
| | Use_sm4: Social media influences your choice of travel destination | |
| | Use_sm5: Social media creates expectations about your travel destination | |
| | Use_sm6: Social media influences you more in choosing a travel destination compared to traditional media (television, radio, newspapers) | |

**Table 2.** *Cont.*

| Variable | Items | Sources (Adapted from) |
|---|---|---|
| Trust in social media for information on health safety (Tr_saf) | Tr_saf1: You trust the messages sent by tourist services suppliers (travel agencies, hotels) on social media on the health safety of a travel destination | Sharma, V.M.; Klein, A. [75] |
| | Tr_saf2: Social media information regarding the health safety of travel destinations will be useful for you in choosing future vacations | |
| | Tr_saf3: You will trust the information on social media regarding the health safety of travel destinations | |
| Intention to purchase a travel destination (Int_t) | Int_t1: You intend to choose a travel destination | McClure, C.; Seock, Y.K. [83] Sharma, V.M.; Klein, A. [75] |
| | Int_t2: You intend to recommend a travel destination | |
| | Int_t3: It is very likely that you will choose a travel destination | |

The construct reliability was evaluated by using two measures: the Cronbach's Alpha coefficient and the composite reliability. As can be seen in Table 3, both Cronbach's Alpha and composite reliability values are higher than 0.7 for all latent variables, therefore demonstrating a good internal consistency reliability of the construct [84].

**Table 3.** Coefficients for the latent variables.

| Variables | Sk_sm | Inv_t | Use_sm | Hlth_s | Tr_sm | Inv_sm | Int_smh | Int_t | Tr_saf |
|---|---|---|---|---|---|---|---|---|---|
| Cronbach's Alpha | 0.854 | 0.755 | 0.939 | 0.827 | 0.840 | 0.886 | 0.904 | 0.891 | 0.843 |
| Composite reliability | 0.897 | 0.845 | 0.952 | 0.897 | 0.926 | 0.917 | 0.921 | 0.932 | 0.905 |
| AVE | 0.639 | 0.578 | 0.768 | 0.743 | 0.862 | 0.688 | 0.566 | 0.821 | 0.762 |
| VIF | 2.043 | 1.549 | 4.038 | 1.566 | 2.031 | 1.853 | 2.841 | 2.259 | 3.161 |

The convergent validity of the construct is guaranteed, the average variance extracted (AVE) for each latent variable being higher than 0.5 [85]. Similarly, the factor loadings should be higher than 0.7 [86], in certain situations being accepted values at least equal to 0.5 [85]. Table 4 displays these values in a shaded background, showing that the majority of the values are higher than 0.7.

**Table 4.** Structure loadings and cross-loadings *.

| | Sk_sm | Inv_t | Use_sm | Hlth_s | Tr_sm | Inv_sm | Int_smh | Int_t | Tr_saf |
|---|---|---|---|---|---|---|---|---|---|
| Sk_sm 1 | 0.882 | 0.480 | 0.559 | 0.185 | 0.381 | 0.438 | 0.520 | 0.485 | 0.440 |
| Sk_sm 2 | 0.883 | 0.503 | 0.573 | 0.258 | 0.408 | 0.419 | 0.548 | 0.485 | 0.461 |
| Sk_sm 3 | 0.757 | 0.271 | 0.503 | 0.157 | 0.314 | 0.379 | 0.389 | 0.420 | 0.416 |
| Sk_sm 4 | 0.619 | 0.340 | 0.330 | 0.191 | 0.236 | 0.230 | 0.343 | 0.271 | 0.307 |
| Sk_sm 5 | 0.826 | 0.402 | 0.625 | 0.183 | 0.390 | 0.456 | 0.495 | 0.513 | 0.464 |
| Inv_t1 | 0.391 | 0.776 | 0.293 | 0.204 | 0.232 | 0.337 | 0.344 | 0.257 | 0.273 |
| Inv_t2 | 0.342 | 0.754 | 0.318 | 0.233 | 0.229 | 0.317 | 0.362 | 0.332 | 0.298 |
| Inv_t3 | 0.490 | 0.801 | 0.434 | 0.271 | 0.341 | 0.355 | 0.483 | 0.397 | 0.380 |
| Inv_t4 | 0.302 | 0.705 | 0.245 | 0.350 | 0.216 | 0.205 | 0.356 | 0.225 | 0.264 |
| Use_sm1 | 0.655 | 0.414 | 0.899 | 0.291 | 0.555 | 0.537 | 0.623 | 0.641 | 0.633 |
| Use_sm2 | 0.543 | 0.407 | 0.855 | 0.346 | 0.533 | 0.552 | 0.635 | 0.604 | 0.644 |
| Use_sm3 | 0.501 | 0.318 | 0.830 | 0.267 | 0.481 | 0.529 | 0.543 | 0.572 | 0.614 |
| Use_sm4 | 0.588 | 0.376 | 0.910 | 0.278 | 0.645 | 0.574 | 0.647 | 0.661 | 0.685 |
| Use_sm5 | 0.593 | 0.383 | 0.894 | 0.292 | 0.614 | 0.559 | 0.642 | 0.665 | 0.652 |
| Use_sm6 | 0.569 | 0.346 | 0.866 | 0.324 | 0.570 | 0.526 | 0.641 | 0.625 | 0.631 |
| Hlth_s1 | 0.245 | 0.309 | 0.340 | 0.853 | 0.366 | 0.193 | 0.380 | 0.281 | 0.470 |
| Hlth_s2 | 0.196 | 0.288 | 0.257 | 0.885 | 0.315 | 0.239 | 0.347 | 0.209 | 0.458 |
| Hlth_s3 | 0.189 | 0.297 | 0.288 | 0.847 | 0.322 | 0.236 | 0.388 | 0.225 | 0.511 |

**Table 4.** *Cont.*

|  | Sk_sm | Inv_t | Use_sm | Hlth_s | Tr_sm | Inv_sm | Int_smh | Int_t | Tr_saf |
|---|---|---|---|---|---|---|---|---|---|
| Tr_sm1 | 0.428 | 0.301 | 0.631 | 0.322 | 0.928 | 0.430 | 0.579 | 0.497 | 0.638 |
| Tr_sm2 | 0.387 | 0.325 | 0.572 | 0.398 | 0.928 | 0.396 | 0.530 | 0.445 | 0.583 |
| Inv_sm1 | 0.364 | 0.310 | 0.520 | 0.189 | 0.361 | 0.862 | 0.473 | 0.430 | 0.410 |
| Inv_sm2 | 0.471 | 0.381 | 0.544 | 0.255 | 0.413 | 0.846 | 0.543 | 0.511 | 0.412 |
| Inv_sm3 | 0.423 | 0.341 | 0.507 | 0.213 | 0.345 | 0.869 | 0.529 | 0.474 | 0.389 |
| Inv_sm4 | 0.340 | 0.362 | 0.425 | 0.220 | 0.314 | 0.758 | 0.493 | 0.365 | 0.341 |
| Inv_sm5 | 0.426 | 0.278 | 0.587 | 0.198 | 0.410 | 0.810 | 0.499 | 0.484 | 0.477 |
| Int_smh1 | 0.397 | 0.420 | 0.439 | 0.417 | 0.444 | 0.372 | 0.755 | 0.403 | 0.496 |
| Int_smh2 | 0.373 | 0.301 | 0.499 | 0.227 | 0.348 | 0.515 | 0.729 | 0.408 | 0.445 |
| Int_smh3 | 0.496 | 0.409 | 0.602 | 0.341 | 0.470 | 0.524 | 0.800 | 0.508 | 0.553 |
| Int_smh4 | 0.488 | 0.361 | 0.594 | 0.247 | 0.458 | 0.533 | 0.773 | 0.558 | 0.534 |
| Int_smh5 | 0.372 | 0.505 | 0.404 | 0.446 | 0.419 | 0.297 | 0.688 | 0.384 | 0.482 |
| Int_smh6 | 0.481 | 0.399 | 0.577 | 0.338 | 0.438 | 0.435 | 0.762 | 0.496 | 0.507 |
| Int_smh7 | 0.481 | 0.335 | 0.581 | 0.313 | 0.538 | 0.518 | 0.740 | 0.523 | 0.479 |
| Int_smh8 | 0.496 | 0.376 | 0.667 | 0.310 | 0.482 | 0.572 | 0.797 | 0.616 | 0.581 |
| Int_smh9 | 0.334 | 0.352 | 0.417 | 0.290 | 0.444 | 0.345 | 0.717 | 0.357 | 0.461 |
| Int_t1 | 0.545 | 0.397 | 0.678 | 0.253 | 0.482 | 0.501 | 0.597 | 0.926 | 0.521 |
| Int_t2 | 0.520 | 0.372 | 0.644 | 0.259 | 0.447 | 0.521 | 0.578 | 0.885 | 0.526 |
| Int_t3 | 0.436 | 0.321 | 0.628 | 0.239 | 0.450 | 0.466 | 0.544 | 0.908 | 0.510 |
| Tr_saf1 | 0.485 | 0.338 | 0.673 | 0.544 | 0.548 | 0.398 | 0.583 | 0.511 | 0.886 |
| Tr_saf2 | 0.477 | 0.333 | 0.707 | 0.436 | 0.585 | 0.476 | 0.607 | 0.563 | 0.908 |
| Tr_saf3 | 0.415 | 0.383 | 0.536 | 0.478 | 0.593 | 0.405 | 0.568 | 0.418 | 0.822 |

\* $p < 0.01$.

The discriminant validity is supported by the fact that the square root of the AVE of each latent variable (Table 5), displayed on the main diagonal, is higher than the correlations of this construct with any other latent variable [87]. At the same time, each loading for the items belonging to the same latent variable is higher than the cross-loadings with other variables, demonstrating that the discriminant validity of the construct is assured [86]. The construct is not affected by collinearity, as the values for the variance inflation factor (VIF) are lower than 5 [86].

**Table 5.** Correlations among latent variables with square roots of the average variance extracted (AVE).

|  | Sk_sm | Inv_t | Use_sm | Hlth_s | Tr_saf | Inv_sm | Int_smh | Int_t | Tr_sm |
|---|---|---|---|---|---|---|---|---|---|
| Sk_sm | 0.800 |  |  |  |  |  |  |  |  |
| Inv_t | 0.505 * | 0.760 |  |  |  |  |  |  |  |
| Use_sm | 0.657 * | 0.427 * | 0.876 |  |  |  |  |  |  |
| Hlth_s | 0.244 * | 0.346 * | 0.342 * | 0.862 |  |  |  |  |  |
| Tr_saf | 0.439 * | 0.337 * | 0.648 * | 0.388 * | 0.928 |  |  |  |  |
| Inv_sm | 0.489 * | 0.402 * | 0.623 * | 0.259 * | 0.445 * | 0.830 |  |  |  |
| Int_smh | 0.581 * | 0.510 * | 0.710 * | 0.431 * | 0.597 * | 0.611 * | 0.752 |  |  |
| Int_t | 0.552 * | 0.401 * | 0.718 * | 0.276 * | 0.507 * | 0.547 * | 0.632 * | 0.906 |  |
| Tr_sm | 0.527 * | 0.401 * | 0.735 * | 0.556 * | 0.658 * | 0.489 * | 0.672 * | 0.572 * | 0.87 |

\* $p < 0.01$.

## 4. Results and Discussions

The designed model was tested using WarpPLS 5.0 software [88,89], with the Structural Equation Modeling procedure being performed. The values of path coefficients (β) and for coefficients of determination ($R^2$) are displayed in Figure 2:

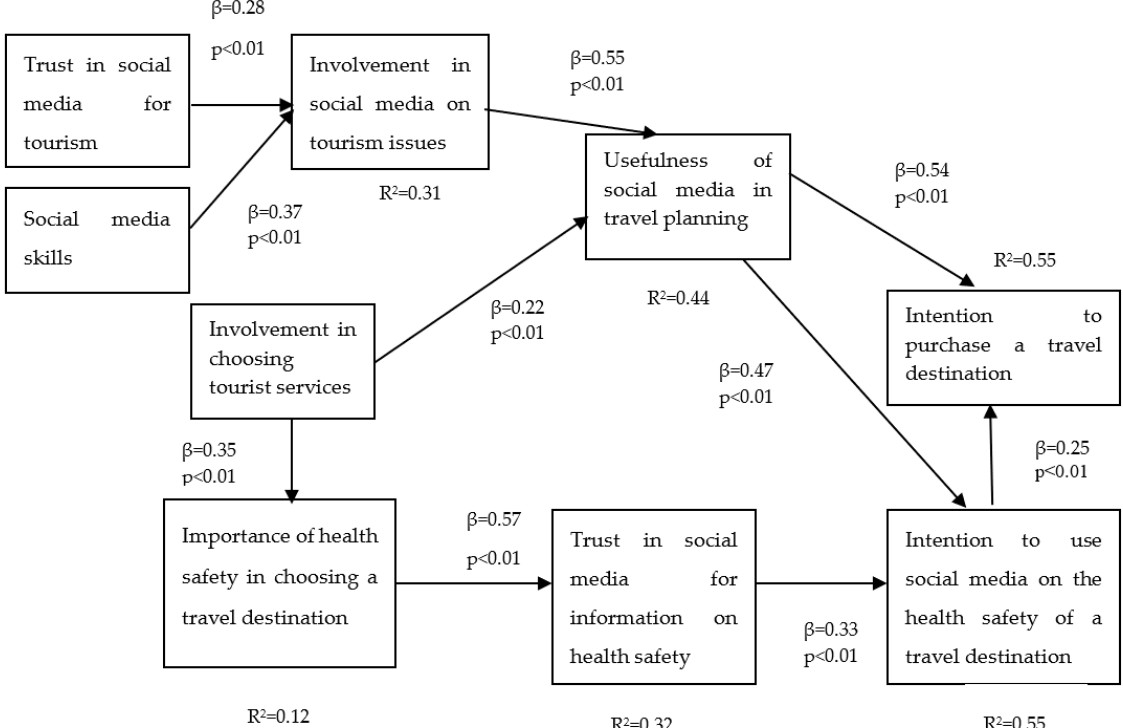

**Figure 2.** The validated model.

The validated model (Figure 2) provides results showing that trust in social media for tourism significantly influences the behavior of sharing travel experience on social media, with the path coefficient being β = 0.28, *p* < 0.01. Thus, the more a person trusts social media for tourism information, the more he/she gets involved in this media by sharing travel information. Consequently, hypothesis H1 is accepted. The persons that trust social media for tourism issues (e.g., tourists' opinions regarding a travel destination) are inclined to become more active in this medium, sharing different kinds of messages (comments, photo content, reviews) on a travel experience.

Social media skills have a significant influence on sharing tourist information through this means, the strength of this influence being rather moderate (β = 0. 37, *p* < 0.01). Consequently, the hypothesis H2 is accepted, the social media experience being positively related to the involvement in this means for acquiring tourist information. The persons that are more highly skilled at using social media (experienced and familiar with the use of this communication medium) are more prone to use this means in sharing tourist information. The value of the coefficient of determination $R^2$ = 0.31 shows that the trust in tourism social media sources and the social media skills account for 31% of the variance in involvement in social media for tourism issues.

The behavior of sharing tourist information through social media significantly influences the usefulness of social media in travel planning. The path coefficient value (β = 0.55, *p* < 0.01) shows this relationship is rather strong, with the hypothesis H3 being accepted. Thus, the individuals that are active on social media in sharing tourist information assign more importance to this means when planning to travel.

The involvement in choosing tourist services (behaving as experienced and exigent tourists, concerned with responsible tourism) has a significant influence on social media usefulness for the travel planning process (β = 0.22, *p* < 0.01), although this influence is rather weak. The consumers that are exigent in the travel decision making process and have experience in this area are more inclined to use social media for choosing a travel destination, meaning hypothesis H4 could be accepted.

The exigent tourists pay more attention to the health safety of a travel destination (stressing more the importance of respecting additional hygiene rules, giving up certain facilities or paying more to

obtain supplementary guarantees in order to reduce the sanitary risk). There is a direct and significant relationship between the variables involvement in choosing tourist services and importance of health safety in selecting a travel destination ($\beta = 0.35$, $p < 0.01$). As a result, hypothesis H5 is accepted.

The importance of health safety in choosing a travel destination has a rather strong influence on the trust in social media that provide information on health safety of tourism destination ($\beta = 0.57$, $p < 0.01$). As people become more aware of the importance of travel sanitary safety, they trust more social media as a source of information (messages on the health safety of a travel destination posted by various entities such as travel agencies, hotels, tourists, etc.), meaning hypothesis H6 was accepted. Furthermore, the coefficient of determination $R^2 = 0.32$ shows that the importance of health safety in choosing a travel destination accounts for 32% of the variance in trust in social media for acquiring information on sanitary safety.

The individuals that trust more social media for information on health safety of travel destinations are more inclined to use this means for acquiring such types of information (intending to use social media instruments like tourism blogs/vlogs, the accounts of friends/acquaintances/influencers, tourism forums, platforms for distributing video/photo content, tourism oriented sites). The relationship between the two variables is rather moderate ($\beta = 0.33$, $p < 0.01$), but significant. Taking into account these arguments, we may conclude that hypothesis H7 is accepted.

Social media usefulness for travel planning has a significant influence ($\beta = 0.47$, $p < 0.01$) on the intention to use social media for searching for information on health safety of tourist destinations. The more importance social media has for tourists, the greater their intention to use it as a source of information on sanitary safety. As a result, hypothesis H8 is accepted. The explanatory capacity of variables trust in social media that provide information on health safety of a travel destination and social media utility for travel planning is quite high ($R^2 = 0.55$), accounting for 55% of the variance of the variable intention to use social media for searching for information on the sanitary safety of a travel destination.

The intention to use social media for searching information on sanitary safety of travel destination rather moderately influences the intention to choose a travel destination ($\beta = 0.25$, $p < 0.01$). The hypothesis H9 is accepted: the individuals that are involved in searching for information regarding the sanitary safety of a travel destination are more inclined to buy tourist services.

Social media usefulness for travel planning has a relatively strong influence on the intention to choose a tourist destination, allowing hypothesis H10 to be accepted ($\beta = 0.54$, $p < 0.01$). As a consequence, the more people assign importance to social media in the travel decision making process, the more likely they are to buy tourist services.

Both variables related to social media (social media usefulness for travel planning and intention to use social media for searching information on health safety of a travel destination) account for 55% of variance of the variable intention to buy a vacation in a specific tourism destination, suggesting that social media is very important for intentions to buy a specific tourist offer.

## 5. Conclusions and Managerial Implications

Social media has become a reliable source of information in the consumer decision making process for a variety of products and services. In the tourism industry, taking into account the intangibility of tourist services, consumers are more inclined to use social platforms to gather information both on a tourism offer [10], and on other aspects that are less easily controlled by tourism suppliers (e.g., the factors of risk a tourist might confront during a trip) [62]. Social media is a very prolific means of communication, as it has the capacity to concentrate and to disseminate information from various categories of providers. The credibility of information issued by social media platforms might vary according to the source status (hotels, travel agencies, tourists, experts, influencers, public authorities), with consumers more likely to trust the information provided by the consumers of tourism services [59].

Considering the travel planning process, as many people have developed skills in using digital technologies, social media gained a major role in helping tourists to choose a travel destination [26].

The present research emphasizes that the more skillful in using online platforms and the more trustful in social media for tourism information people are, the greater their use of social media, sharing comments, photos, videos and reviews related to visited destinations. Thus, the role of tourists might be transformed towards one that involves spreading authentic travel experiences. One implication could be that tourist services suppliers might benefit from the amplitude of the disseminating information process regarding their offers. At the same time, they become aware of the consequences generated by negative experiences of tourists and try to avoid them by offering higher quality services and honestly dealing with the forms of dissatisfaction expressed on social media (e.g., by answering negative comments, giving additional explanations).

The perceived usefulness of social media in the travel planning process (to plan a holiday, practice a responsible form of tourism, choose a tourism destination or generate expectations in order to evaluate a travel experience) is influenced, according to the model, by the individuals' involvement in choosing tourist services and using social media for travel information. Therefore, people that are more concerned with tourist services and are more active on social media perceive this means to be more useful in travel planning and are more willing to use it, as compared to traditional media (television, radio, newspapers). Tourism suppliers might take advantage of this tendency through the intensification of their communication actions in social media, which could aim to attract the exigent tourists that might be more resistant to other persuasion methods.

Health safety has become an important criterion for selecting a travel destination, especially in the actual context, induced by SARS-CoV-2 pandemic. According to the present research, tourists are more concerned about their health and their decision making process for buying tourism services has begun to favor destinations that are considered more safe, to the detriment of other supply characteristics. To reduce health risks, tourists are willing to pay extra money, give up certain facilities and choose destinations where the supplementary hygiene rules are visible. All these preoccupations, combined with the significant increase of online communication determined by social distancing rules, have favored the role of social media in providing up to date information related to the potential health risks during travel. The present research suggests that the exigent tourists are more prone to emphasize health safety when choosing a travel destination. In order to attract this segment of tourists, tourism suppliers should communicate the actions implemented to reduce health risks (imposed by the public health authorities, the professional associations, the quality assurance authorities, etc.) through particular media, with social platforms being one of the most highly recommended methods. People that assign more importance to health safety when choosing a travel destination trust more social media as a source of information regarding sanitary safety than other sources. An important consequence of this is that the credibility conferred to information on health safety communicated through social media by different actors from the tourism industry (tourism organizations, public authorities, tourists that have traveled to a certain destination) influences the intention to use this means on the above mentioned purpose. Moreover, tourists that consider social media to be useful in the travel planning process are more inclined to use social platforms for information on the health safety of a travel destination. Thus, people that are active in social media are interested in searching for health safety information on tourism blogs/vlogs, social media accounts of friends/acquaintances, tourists' forums, official sites of tourism services suppliers or tourism oriented sites (e.g., TripAdvisor). One managerial implication of these findings targets the tourism suppliers that should share updated and complete information on sanitary safety of a destination, and become more involved in interactive communication with tourists that are interested in obtaining additional information. The trust in the sanitary safety of a destination might increase its attractivity and confer on it an outstanding competitive advantage [51].

According to the model validated in this paper, the intentions to buy a travel destination are influenced by the perceived usefulness of social media in the travel planning process and the intention to use social media regarding the health safety of a travel destination. The coefficient of determination

($R^2 = 0.55$) shows that buying intentions are explained in great measure by the two variables, outlining the major role social media has in increasing the purchasing intentions towards a package tour.

The model developed in this paper is useful for various tourism decision making organizations (hotels, travel agencies, ministries of tourism) in managing their social media marketing actions. Especially during sanitary crises, these organizations have to target social media users and to disseminate reliable online information on the health safety of tourism destinations. In this manner, they will increase the trust in social media information on tourism, stimulating intentions to buy various tourism packages.

Other categories of stakeholders are also targeted. For example, the public health authorities may use social media to increase awareness of some health risk situations, to educate tourists to respect the sanitary rules and to provide information about restrictions in different areas. Consumers' organizations could communicate on the facilities that tourists have for searching for additional information on health safety or medical assistance in certain destinations.

Further research may evaluate the trust conferred to different tourism information shared on social media (price, accommodation, health safety) while considering the status of the contributor, which could be formal (e.g., tourism suppliers) or informal (e.g., tourist forums, tourist accounts, blogs, vlogs or tourist podcasts). Furthermore, the model can include other variables as characteristics of travel destinations or motivations to travel.

The papers limitations involve the period when the data was collected. The threat caused by SARS-CoV-2 pandemic might have caused the subjects to overestimate the role of health safety. Moreover, this context might create uncertainties regarding the travel planning that could affect their buying intentions for a tourism destination. Furthermore, a qualitative approach could add supplementary value to the results of the quantitative method. Thus, in a future study, qualitative research might be used to reveal additional coordinates of tourist behavior when selecting a travel destination under sanitary risk conditions.

**Author Contributions:** Conceptualization, C.-E.Ț., D.-M.V. and C.-E.N.; methodology, C.-E.Ț., D.-M.V. and C.-E.N.; formal analysis, C.-E.Ț. and D.-M.V.; investigation, C.-E.Ț. and D.-M.V.; data curation, D.-M.V.; writing—original draft preparation, C.-E.Ț., D.-M.V.; writing—review and editing, C.-E.Ț., D.-M.V. and C.-E.N.; supervision, D.-M.V.; funding acquisition, C.-E.N. All authors have read and agreed to the published version of the manuscript.

**Funding:** This research received no external funding.

**Conflicts of Interest:** The authors declare no conflict of interest.

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
