# Peer review of "The Role of Social Media in Health Safety Evaluation of a Tourism Destination throughout the Travel Planning Process"

_sustainability, doi:10.3390/su12166661_

Round 1
Reviewer 1 Report
Lines and following: H6, 7,8,10 no comments and explanations, while for all the other hypotheses the authors provide them. Therefore they should also be inserted for the four aforementioned.
Line 198 Figure 1: in the proposed model the passage of the H4 "Involvement in choosing" is not clear to me; it would seem disconnected from the other parts of the model.
As the model is very technical, I suggest adding further explanations of the results and their applicability in the conclusions.
Author Response
We would like to warmly thank you for your careful and thorough reading of the manuscript and for the comments and constructive suggestions.
Point 1: Lines and following: H6, 7,8,10 no comments and explanations, while for all the other hypotheses the authors provide them. Therefore they should also be inserted for the four aforementioned.
Response 1: Initially, we considered these 4 hypotheses to be much inter-related and we demonstrated all 4 in the previous paragraph. At your suggestion, we separated the information into two distinct paragraphs, as we considered it to justify the hypotheses. Thus, hypotheses 6 and 7 are demonstrated separately from hypotheses 8 and 9.
Point 2: Line 198 Figure 1: in the proposed model the passage of the H4 "Involvement in choosing" is not clear to me; it would seem disconnected from the other parts of the model.
Response 2: There was a word display error. Two words were hidden: tourist services. These have been displayed.
Point 3: As the model is very technical, I suggest adding further explanations of the results and their applicability in the conclusions.
Response 3: In the Results and discussion section we added more explanations; also, in the conclusions section.
The explanations are marked in red and can be found in the lines (in our editing format): 264-266; 270-271; 280-281; 285-287; 293-295; 300-303; 390-395.

Reviewer 2 Report
Well written article and the topic is very very actual. It was very interesting to read the article. From the academic point of view everything is correct and well done.
The topic and aim of this research was to identify the role of social media in evaluating the attractiveness of a tourist destination with a special emphasis on the health safety of the destination. This topic is much more actual and important today as it was last year or before that. Because of the Covid-19 pandemic the health safety has become the most important criterion of selecting a travel destination. According to the present research tourists need to reduce the health risk and willing to pay extra money to choose safe destinations. When choosing a travel destination, the greater the attention tourists pay to social media. Since the data from this study were collected before the Covid-19 pandemic, it would be very important to conduct a re-stydy to see if and how the pandemic has affected the use of social media in choosing the travel destination.
Author Response
Thank you very much for your time and your appreciation of our research results. As well, thank you for your suggestion. As the research was conducted in the middle of the pandemic, the data being collected during May, it would be really useful to resume the research after the situation returns to normal.
Thank you for support!

Reviewer 3 Report
The paper explores with clarity the role of social media in shaping tourist attitudes, notably with reference to health issues. Although it is not new in terms of theoretical frameworks and methodologies, it is valuable in terms of structure development and clarity. Anyway, some limitations should be highlighted in the article, such as the too much quantitative-oriented perspective which does not take into account the relevant role of qualitative methods in scrutinizing such a topic.
Author Response
First of all, we wish to thank you for your constructive comments. As you suggest, we tried to highlight the limitation brought out by an exclusive quantitative-oriented perspective. In this way, we added, at the end of the section about limitations, this paragraph: <Furthermore, a qualitative approach could add a supplementary value to the results of quantitative method. Thus, in a future study, a qualitative research might be used to reveal additional coordinates of tourist behaviour in selecting a travel destination under sanitary risk conditions.>
The paper was also revised in terms of English accuracy.
